# SIA-OVD: Shape-Invariant Adapter for Bridging the Image-Region Gap in Open-Vocabulary Detection

Zishuo Wang
Wangxuan Institute of Computer Technology, Peking University
Beijing, China
wangzishuo@pku.edu.cn

Wenhao Zhou
School of Intelligence Science and Technology, University of Science and Technology Beijing
Beijing, China
M202320876@xs.ustb.edu.cn

Jinglin Xu
School of Intelligence Science and Technology, University of Science and Technology Beijing
Beijing, China
xujinglinlove@gmail.com

Yuxin Peng*
Wangxuan Institute of Computer Technology, Peking University
Beijing, China
pengyuxin@pku.edu.cn

## Abstract

Open-vocabulary detection (OVD) aims to detect novel objects without instance-level annotations to achieve open-world object detection at a lower cost. Existing OVD methods mainly rely on the powerful open-vocabulary image-text alignment capability of Vision-Language Pretrained Models (VLM) such as CLIP. However, CLIP is trained on image-text pairs and lacks the perceptual ability for local regions within an image, resulting in the gap between image and region representations. Directly using CLIP for OVD causes inaccurate region classification. We find the image-region gap is primarily caused by the deformation of region feature maps during region of interest (RoI) extraction. To mitigate the inaccurate region classification in OVD, we propose a new Shape-Invariant Adapter named SIA-OVD to bridge the image-region gap in the OVD task. SIA-OVD learns a set of feature adapters for regions with different shapes and designs a new adapter allocation mechanism to select the optimal adapter for each region. The adapted region representations can align better with text representations learned by CLIP. Extensive experiments demonstrate that SIA-OVD effectively improves the classification accuracy for regions by addressing the gap between images and regions caused by shape deformation. SIA-OVD achieves substantial improvements over representative methods on the COCO-OVD benchmark. The code is available at https://github.com/PKU-ICST-MIPL/SIA-OVD_ACMMM2024.

## CCS Concepts

• **Computing methodologies** → **Artificial intelligence**; **Computer vision**; **Computer vision problems**; **Object detection**;

*Corresponding author.

## Keywords

Open-Vocabulary Detection, Shape-Invariant, Image-Region Gap, Feature Adapter

**ACM Reference Format:**
Zishuo Wang, Wenhao Zhou, Jinglin Xu, and Yuxin Peng. 2024. SIA-OVD: Shape-Invariant Adapter for Bridging the Image-Region Gap in Open-Vocabulary Detection. In *Proceedings of the 32nd ACM International Conference on Multimedia (MM '24), October 28-November 1, 2024, Melbourne, VIC, Australia.* ACM, New York, NY, USA, 9 pages. https://doi.org/10.1145/3664647.3680642

## 1 Introduction

Object detection is a fundamental visual task that requires a massive amount of annotated bounding boxes for training, which limits the ability of traditional object detectors to cope with the continual emergence of novel objects. Therefore, OVR-CNN [37] proposes an Open-Vocabulary Detection (OVD) setting, which trains object detectors without bounding box annotations for target objects. Subsequently, models are tested on object detection for target categories, thereby achieving the capability to detect newly introduced objects without the need for additional annotations. The OVD framework enables versatile real-world applications.

Most OVD methods leverage the powerful visual-language model CLIP [24] pre-trained on large image-text datasets to achieve open-vocabulary detection capabilities. CLIP embeds texts and images into a common feature space and shows remarkable zero-shot ability in open-vocabulary vision tasks. Mainstream OVD methods divide detection into localization and classification, and mainly focus on the classification stage. In this stage, CLIP is treated as the open-vocabulary classifier. The CLIP text encoder calculates textual embeddings of each category, and the CLIP image encoder calculates visual embeddings of each region proposal. Then, the final OVD results are gained from the cosine similarity of the category embeddings and region proposal embeddings.

However, applying CLIP to OVD may raise a fundamental challenge: CLIP is pre-trained on image-text pairs, and the CLIP image encoder can effectively extract features from entire images, while OVD requires to capture visual features of regions containing objects of interest within an image. To this end, previous efforts [1, 2, 11, 20, 34] employed RoIAlign [10] to crop regions from entire

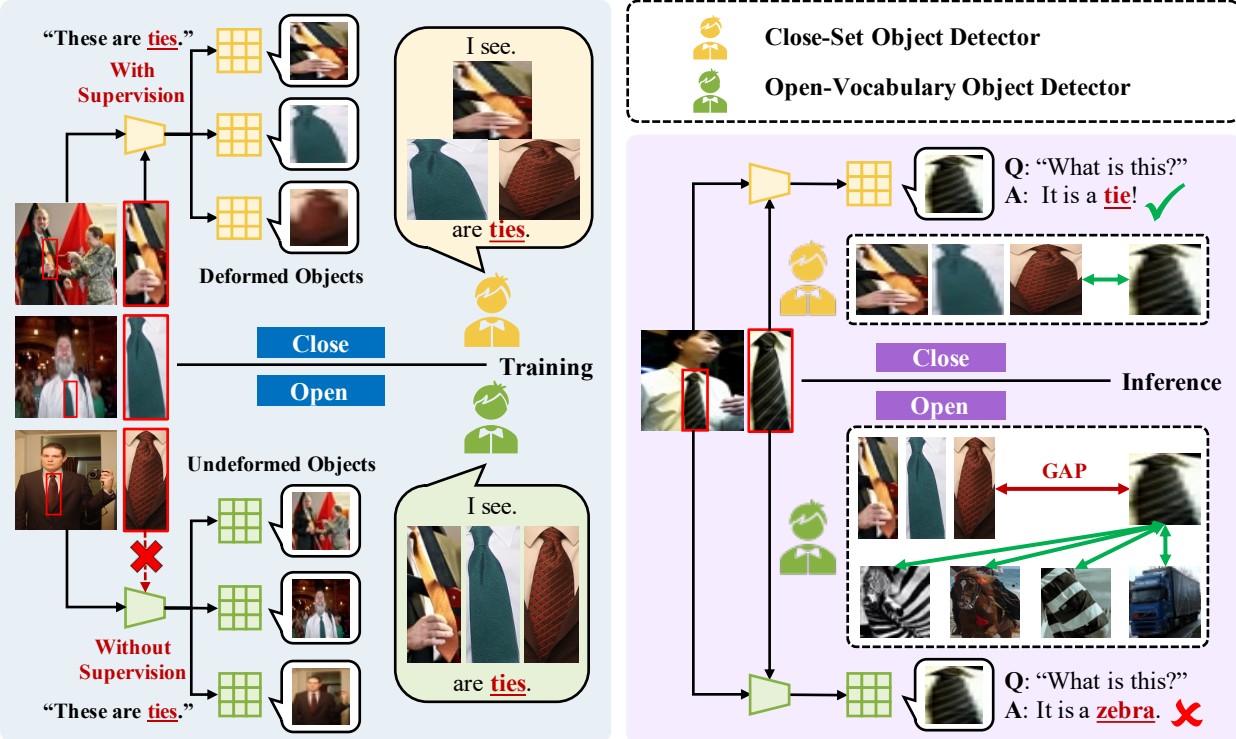

Figure 1: Comparison between Close-Set Object Detection and Open-Vocabulary Object Detection. The Close-Set detector learns object knowledge from instance-level supervision, where RoIAlign deforms the regions of objects. The OVD detector learns from image-level supervision, where the regions of objects keep the original shape. This difference causes the gap between the image and region in OVD, especially for deformed object regions.

images. However, the image-text alignment via CLIP was corrupted after using RoIAlign [40], resulting in a gap between the image and region. Such a gap leads to poor classification accuracy for novel objects. Recent works have further attempted to eliminate this gap to improve the accuracy of region classification by region-level knowledge-distillation[6, 9, 21], region-text pre-training/fine-tuning [16, 22, 32, 40], and visual/textual prompting [5, 7, 33]. However, these methods fail to find the fundamental reason for this gap. In this work, we argue that **object shape deformation brought by RoIAlign is the main cause of the image-region gap**. Figure 1 intuitively illustrates how the image-region gap is caused by object shape deformation. **This gap has brought an obstacle to transferring the open-vocabulary ability of VLMs to dense prediction tasks such as object detection and semantic segmentation**. Specifically, this obstacle is reflected in the low classification accuracy for regions in the OVD task. Therefore, addressing this issue in the OVD task is urgent.

In this work, we propose a new Shape-Invariant Adapter (SIA) to bridge the gap between the image and region caused by the object shape deformation. Specifically, SIA consists of a set of independent adapters that transform the object region's features into shape-invariant region features to mitigate the image-region gap. Each SIA adapter is a lightweight bottleneck network to prevent the potential over-fitting problem on base categories since the shapes of object regions are a long-tailed distribution. In addition, we design

a new adapter allocation mechanism in SIA to adaptively adjust the weights of different adapters according to objects' shapes. Thus, each adapter only needs to cope with regions with similar shapes, which reduces the image-region gap caused by shape variance. So far, SIA can better align the shape-invariant region features with text features to improve region classification accuracy in the OVD task. It is worth mentioning that SIA allows us to use the frozen CLIP image encoder as the backbone of our OVD model without fine-tuning its parameters, which brings the benefits of CLIP's open-vocabulary knowledge directly into the OVD task, especially in scenarios where traditional object detection models struggle with novel objects. This incorporation will lead to more robust and adaptable detection systems.

The contributions of this work are summarized as follows:

- We propose a new Shape-Invariant Adapter (SIA) by exploring the fundamental cause of the gap between image and local region in the OVD task to bridge this gap and improve region classification accuracy effectively.
- We design a new Adapter Allocation Mechanism that adjusts the weights for adapters in SIA according to objects' shapes, helping mitigate the image-region gap caused by shape variance.
- We evaluate our SIA on the COCO-OVD benchmark, which achieves improvements over representative methods on both open-vocabulary detection and region classification.

## 2 Related Works

### 2.1 Open-Vocabulary Detection

**Knowledge-Distillation based OVD** methods tend to distill the open-vocabulary knowledge of vision-language pre-trained models (VLM), e.g., CLIP, into traditional close-set object detectors. ViLD [9] designs a distillation loss between CLIP image embeddings and image features obtained by the detector's backbone to transfer CLIP's visual capabilities to the object detector and leverages the CLIP text encoder for efficient classification within the object detection framework. HierKD [21] proposes a global-level knowledge distillation method and combines it with instance-level knowledge distillation to simultaneously learn base and target categories. OADP [30] proposes an Object-Aware Distillation Pyramid to mitigate the information destruction and inefficient knowledge transfer in distillation-based OVD methods. MM-OVOD [13] leverages the powerful generative ability of LLMs to enrich the textual descriptions of unseen categories. Then, it constructs a multi-modal classifier by combining image exemplars and textual descriptions, which achieves higher classification performance for novel categories. VL-PLM [39] and SAS-Det [38] introduce a self-training strategy into OVD, using CLIP to generate pseudo labels for novel categories based on the region proposals predicted by a class-agnostic proposal generator. Then a close-set detector is trained on the pseudo-labels for novel categories generated by CLIP. SIC-CADS [6] refines the object classification scores of an existing OVD model by leveraging global knowledge distilled from a frozen CLIP model. These knowledge-distillation-based OVD methods attempt to equip close-set detectors with the ability to detect open-vocabulary words. In contrast, we utilize the frozen CLIP image encoder as the backbone of our detector, which directly incorporates CLIP's open-vocabulary knowledge into the OVD task.

**Region-Text Pretraining based OVD** methods tend to fine-tune visual encoders on region-text pairs to enhance the alignment between region and text features for more precise object classification. RegionCLIP [40] presents a two-stage training strategy, which involves pretraining by aligning region-text pairs, followed by transferring the acquired knowledge to an object detector. GLIP [16] unifies the formulation of object detection and phrase grounding, which learns the correlation between regions and sub-words. OWL-ViT [22] first conducts image-level contrastive pretraining and then transfers to object-level with a bipartite matching loss. RO-ViT [14] employs region positional embeddings that are randomly cropped and resized instead of using positional embeddings for the entire image to bridge the image-region gap. Edadet [26] proposes Early Dense Alignment to improve the generalization of local semantics and maintain the local fine-grained semantics. CLIM [32] aggregates multiple images and treats each image as a pseudo-region to form region-text pairs for fine-tuning CLIP. Moreover, it can seamlessly integrate with other OVD models as a versatile framework. YOLO-World [4] is pre-trained on large-scale datasets, which maintains powerful zero-shot recognition capabilities while still having efficient inference speed. Some of these methods suffer from the noisy pseudo-region-text pairs [12]. Additionally, fine-tuning the parameters of CLIP is likely to destroy its open-vocabulary ability to generalize to novel objects. In contrast, we keep the CLIP image encoder frozen to maintain CLIP's generalization ability.

**Prompt-Learning based OVD** methods usually keep the CLIP image encoder frozen and introduce learnable prompts to adapt CLIP to the object detection task. DetPro [5] learns context tokens as input to the text encoder. It learns different prompts for proposals of different IoU with ground-truth boxes to address the variances among proposals. PromptDet [7] incorporates learnable prompt vectors into textual input, which are unrelated to categories, enabling generalization from base classes to target classes. Prompt-OVD [27] propose a prompt-based decoding module, in which both the image embedding and text embedding from CLIP are regarded as class prompts for each self-attention module of the decoder. CORA [33] utilizes ground truth bounding boxes from base classes to train position-embedding modules within the RN50 backbone, aiming to adapt CLIP to the OVD task better. According to the section 2.1, these prompt-learning-based OVD methods learn fixed prompts for all samples, which may limit their generalization ability for unseen categories. In contrast, we learn conditional adapters for different samples for better generalization performance.

### 2.2 Multi-Modal Prompt Tuning

Large-scale vision-language pre-trained model (VLM), e.g., CLIP, demonstrates impressive zero-shot performance across various downstream tasks [17, 18, 29, 35]. Adapting VLMs to a specific task commonly involves two approaches: fine-tuning or prompt tuning. Recently, the focus of research has gradually shifted towards prompt tuning, which keeps VLM encoders frozen and learns additional prompts as input to VLMs. CoOp [42] replaces the manual template of textual prompts with learnable context tokens, which are trained with all categories. However, such fixed context tokens result in poor zero-shot capabilities and generalization to unseen categories, so CoOp is not compatible with the OVD task. CoCoOp [41] integrates image features into the learning process by adding a lightweight meta-net to generate instance-conditional textual tokens to mitigate this issue. CoCoOp improves the open-vocabulary ability, but the memory consumption and computational cost are extremely high because the meta-net does not support batch processing. UPT [36] developed a modality-agnostic prompt tuning method, leveraging a lightweight transformer layer to concurrently fine-tune both the visual and textual branches. CLIP-Adapter [8] adapts the original clip model to downstream tasks by adding trainable linear layers after the vision encoder and text encoder. CLIP-Adapter achieves higher efficiency because the adapters are inserted after encoders, and the back-propagate paths are much shorter. FG-VPL [28] explores the potential for fine-grained image recognition of CLIP by introducing fine-grained visual prompts. In this paper, we follow these prompt tuning methods with frozen backbones to maintain the generalization ability of CLIP.

## 3 Methodology

### 3.1 Overview

Open-vocabulary detection (OVD) aims to train an object detector $\mathcal{D}$ to detect objects of novel categories $C_N$ without the instance-level annotations for them. During the training phase, only instance-level annotations for objects of base categories $C_B$ and the pre-trained CLIP model are available. During the inference phase, the detector is tested on a set of novel categories $C_N$, where $C_B \cap C_V = \emptyset$.

                                                                           Zishuo Wang, Wenhao Zhou, Jinglin Xu and Yuxin Peng

**SIA-OVD: Shape-Invariant Adapter for Bridging the Image-Region Gap in Open-Vocabulary Detection**

**Figure 2: Overview of the SIA-OVD framework. It takes image and prompt templates filled in with class names as input and outputs the bounding boxes of objects in the whole image along with prediction classification.**

Extra training datasets of image-text pairs (e.g., COCO Captions [3] and CC3M [25]) are also available, but in this work, we do not utilize any extra datasets to train our model.

Specifically, our open-vocabulary object detection process is divided into two branches: object localization and region classification. We mainly focus on improving the accuracy of region classification in this work. For object localization, we use DAB-DETR [19] as the localizer, which learns object queries with bounding boxes as input. DAB-DETR encodes the feature map output by the $3^{rd}$ layer of the CLIP image encoder and decodes the outputs of the bounding boxes of detected objects in both base and novel categories. For region classification, given a region proposal of 4 dimensions $(x_i, y_i, h_i, w_i)$ with its confidence score $conf_i$ decoded from DAB-DETR, we utilize RoIAlign [10] to obtain the region feature from CLIP's image feature map. Then, we feed the region feature and the bounding box into the Shape-Invariant Adapter (SIA). SIA generates the weights for different adapters according to the shape of the bounding boxes and transforms the region features into shape-invariant ones that are better aligned with text embeddings. We perform region classification by calculating the cosine similarities between region embeddings and text embeddings of all categories. The overall framework is illustrated in Figure 2.

### 3.2 Shape-Invariant Adapter (SIA)

**Structure of SIA.** In this section, we introduce our Shape-Invariant Adapter illustrated in Figure 3. To mitigate the image-region gap in OVD, SIA aims to learn a mapping from the region feature space obtained by RoIAlign to the common space, which is aligned with CLIP visual and textual features. We are inspired by the fact that CoOp [42] learns fixed context tokens for all samples and performs poorly on unseen categories, while CoCoOp [41] learns conditional context tokens based on each sample and achieves better performance on unseen categories. Therefore, we try to learn conditional feature transformation based on the shapes of region proposals.

Luckily, the shape information of a region is much simpler than its semantic information. We do not have to rely on the generative network of CoCoOp to generate conditional context, which results

in a high computational cost. Specifically, SIA keeps a set of $N$ independent adapters $\{Adapter_j, j = 1, ..., N\}$. Each adapter consists of two fully connected layers $\mathbf{W}_1^j$ and $\mathbf{W}_2^j$, a ReLU activation layer, and a residual factor $\lambda$. When the $i^{th}$ region feature $f_r^i \in \mathbb{R}^{D \times 1}$ is input to SIA, we obtain the adapted region embedding $\hat{f}_r^i$ through a residual connection:

$$Adapter_j(f_r^i) = \text{ReLU}(f_r^i \mathbf{W}_1^j)\mathbf{W}_2^j, \tag{1}$$

$$\hat{f}_r^{i,j} = \lambda * Adapter_j(f_r^i) + (1 - \lambda) * f_r^i. \tag{2}$$

For the $i^{th}$ region proposal $\mathcal{R}_i$, SIA outputs $N$ adapted region features:

$$F_r^i = (\hat{f}_r^{i,1}, ..., \hat{f}_r^{i,N}) \in \mathbb{R}^{D \times N}. \tag{3}$$

**Adapter Allocation Mechanism.** To mitigate the image-region gap caused by shape variance, we design an Adapter Allocation Mechanism, which generates weights for the $N$ adapter in SIA so that each adapter only needs to cope with regions with similar shapes. We quantize the region shapes as aspect ratios (height/width). Then we manually partition the real number axis of different aspect ratios into $N$ discrete intervals by $N + 1$ points $\{s_0, s_1, ..., s_N\}$ ($s_0 = 0$ and $s_N \rightarrow +\infty$). Given a region proposal $\mathcal{R}_i$ with a height of $h_i$ and a width of $w_i$ from the object query of DETR, we compute the one-hot allocation weight according to its shape $h_i/w_i$:

$$Y_i = (y^{(1)}, ..., y^{(N)}) \in \{0, 1\}^{1 \times N}$$
$$with\ y^{(k)} = \mathbf{1}[s_{k-1} < h_i/w_i \le s_k], 1 \le k \le N. \tag{4}$$

Given the adapter allocation weights $Y_i$ for the $i^{th}$ region proposal and the region features adapted by all adapters in SIA $F_r^i$, we multiply the region features by the weights to select the optimal adapter:

$$\beta_r^i = F_r^i \times Y_i^{\text{T}} \in \mathbb{R}^{D \times 1}. \tag{5}$$

## Shape-Invariant Adapter (SIA)

Figure 3: Illustration of Shape-Invariant Adapter.

Finally, to perform classification, the adapted region feature $\beta_r^i$ is multiplied by the CLIP text embeddings for each category name. The text embeddings are denoted as:

$$\{\alpha_t^k, k = 1, ..., K\} \in \mathbb{R}^{D \times K}, \tag{6}$$

where $K$ is the number of categories. The probability that region $\mathcal{R}_i$ belongs to category $k$ is denoted as:

$$p_i^k = \frac{\exp\left((\beta_r^i)^{\mathrm{T}}(\alpha_t^k)/\tau\right)}{\sum_{j=1}^K \exp\left((\beta_r^i)^{\mathrm{T}}(\alpha_t^j)/\tau\right)} \tag{7}$$

### 3.3 Training and Inference

**Training.** We adopt a two-stage training process. In the first stage, we freeze all parameters except the adapters in SIA. For a training sample $\mathcal{R}_i$, the ground-truth bounding boxes are input to the classification branch, and we optimize SIA with Cross-Entropy Loss:

$$\mathcal{L}_{cls} = -\frac{1}{K} \sum_{k=1}^K \log p_i^k \tag{8}$$

Then we keep the classification branch frozen and train the transformer encoder and decoder in the localization branch. Our localizer shares the same DAB-DETR structure as CORA [33], so we utilize the same training object.

**Inference.** During inference, for the $i^{th}$ object query, the bounding box is refined by the transformer decoder and then fed to SIA for classification. We obtain a localization score $score_l$ and a classification score $score_c$ respectively for the region proposal. The classification score denoted the probability that this proposal belongs to a certain category:

$$score_c = \max_{k=1,...,K} p_i^k. \tag{9}$$

The localization score is from the anchor pre-matching strategy of CORA. We multiply $score_c$ with $score_l$ as the confidence of this region proposal:

$$score_{box} = score_c \cdot score_l \tag{10}$$

## 4 Experiments

### 4.1 Dataset and Evaluation Metrics

**COCO-OVD Benchmark.** COCO-OVD benchmark is proposed by OVR-CNN [37], which splits the COCO categories into 48 base categories and 17 novel categories. Under the OVD setting, object detectors are trained on the bounding box annotations of 48 base categories in the training set. Detectors are evaluated on the validation set, which contains 28,538 instances of base categories and 4,614 instances of novel categories.

**OV-LVIS Benchmark.** Following [40], we also conduct experiments on the OV-LVIS benchmark, which has 1023 categories in total. The model is trained on the common and frequent categories and tested on the rare categories.

**Evaluation Protocol.** We mainly evaluate the model under the generalized setting, which demands the model to detect objects of both base and novel categories at the same time. We take AP50 (average precision with an IoU threshold of 0.5) as the evaluation metric and report the metric on base categories and novel categories separately. Additionally, to directly reflect the classification performance for objects, we also calculate AP50 and box classification accuracy (denoted as Acc.) with a ground-truth bounding box as input.

### 4.2 Implementation Details

**Model Architecture.** The architecture of the object localizer is identical to CORA, which is a DAB-DETR [19] consisting of a 3-layer transformer encoder, a 6-layer transformer decoder, and 1,000 learnable object queries as input. We utilize the 80 prompt

**Table 1: Experimental results on the COCO-OVD benchmark.**

| Method | Publication | Backbone | Supervision | | | AP50 (Generalized) | | |
| | | | Extra Dataset | Pretrained Model | Require Novel Class | Novel | Base | All |
|---|---|---|---|---|---|---|---|---|
| OVR-CNN [37] | CVPR 21 | RN50 | COCO-Captions | BERT-base | No | 22.8 | 46.0 | 39.9 |
| Detic [43] | ECCV 22 | RN50 | COCO Captions | CLIP (Text Encoder) | No | 27.8 | 47.1 | 45.0 |
| RegionCLIP [40] | CVPR 22 | RN50 | CC3M | CLIP (RN50) | No | 31.4 | 57.1 | 50.4 |
| VL-PLM [39] | ECCV 22 | RN50 | - | CLIP (RN50) | Yes | 34.4 | 60.2 | 53.5 |
| PromptDet [7] | ECCV 22 | RN50 | LAION-400M | CLIP (Text Encoder) | Yes | 26.6 | - | 50.6 |
| F-VLM [15] | ICLR 23 | RN50 | - | CLIP (RN50) | No | 28.0 | - | 39.6 |
| BARON [31] | CVPR 23 | RN50 | - | CLIP (RN50) | No | 34.0 | 60.4 | 53.5 |
| CORA [33] | CVPR 23 | RN50 | - | CLIP (RN50) | No | 35.1 | 35.5 | 35.4 |
| **SIA (Ours)** | - | RN50 | - | CLIP (RN50) | No | **35.5** | 40.3 | 39.3 |
| RegionCLIP [40] | CVPR 22 | RN50x4 | CC3M | CLIP (RN50x4) | No | 39.3 | 61.6 | 55.7 |
| CORA [33] | CVPR 23 | RN50x4 | - | CLIP (RN50x4) | No | 41.7 | 44.5 | 43.8 |
| ProxyDet [12] | AAAI 24 | RN50x4 | COCO Captions | - | No | 30.4 | 52.6 | 46.8 |
| **SIA (Ours)** | - | RN50x4 | - | CLIP (RN50x4) | No | **41.9** | 48.8 | 40.5 |

**Table 2: AP50 and Classification Accuracy on ground-truth bounding boxes from the COCO-OVD dataset.**

| Methods | Backbone | AP50 | | Acc |
| | | Novel | Base | |
|---|---|---|---|---|
| CLIP [24] | RN50 | 58.2 | 58.9 | 44.0 |
| CoOp [42] | RN50 | 64.4 | 75.7 | - |
| CLIP-Adapter [8] | RN50 | 63.0 | **80.6** | - |
| CORA [33] | RN50 | 65.1 | 70.0 | 74.3 |
| **SIA (Ours)** | RN50 | **68.6** | 79.5 | **81.3** |
| CLIP [24] | RN50x4 | 63.9 | 62.7 | 51.9 |
| CORA [33] | RN50x4 | 74.1 | 76.0 | 78.3 |
| **SIA (Ours)** | RN50x4 | **75.8** | **85.4** | **83.0** |

templates provided by CLIP to calculate the text embeddings for each category.

**Training & Hyperparameters.** As described in section 3.2, we adopt a two-stage training process. First, we take the ground-truth bounding boxes as input, freeze the backbone, and only update the parameter of $N$ adapters. According to the ablation studies in section 4.3, our model achieves the best performance when $N$ equals to 10. We train the adapters for 5 epochs with a base learning rate of $10^{-4}$, which decays after 4 epochs by a factor of 0.1. In the second stage, we freeze the adapters and update the localizer for 35 epochs with a learning rate of $10^{-4}$. The first training stage takes 4 GPUs with a batch size set to 16. The second training stage is the same as CORA does, which takes 8 GPUs with batch size set to 16.

## 4.3 Results and Analysis

**Comparison with State-of-the-Art.** Table 1 presents our main results on the COCO-OVD benchmark. The compared OVD methods include knowledge-distillation based methods (F-VLM [15], SIC-CADS [6]), region-text pretraining based methods (RegionCLIP [40], BARON [31], CLIM [32]), pseudo-labeling methods (VL-PLM [39]) and prompt-learning methods (PromptDet [7], DetPro [5], CORA [33]). We do not compare SIA with SIC-CADS and CLIM, which are attached to an existing OVD model.

To conduct a fair comparison, we separately compare the aforementioned methods with different visual backbones: RN50 and RN50x4. For each method, we list its supervision from three aspects: Extra Dataset (data beyond the instance-level annotation for base categories), Pretrained Model (the CLIP image encoder and text encoder), and Require Novel Class (whether the names of novel categories is needed during training).

Among these baselines, SIA outperforms CORA [33] by 0.4 AP50 on novel categories with RN50 backbone, and 0.3 AP50 with RN50x4 backbone. SIA only utilizes the pre-trained CLIP model and instance-level annotations for base categories, without the need for any extra dataset. Moreover, during the training phase, novel class names are not required either, which makes it more convenient to train the model and easier to expand to unseen categories.

**Experiments on Ground-Truth Bounding Boxes.** To verify the effectiveness of SIA on region classification, we take the ground-truth bounding boxes for novel categories as input and calculate AP50 and average classification accuracy. The results are shown in Table 2. We compare SIA with CLIP [24], CoOp [42], CLIP-Adapter [8], RegionCLIP [40], and CORA [33]. Due to the noises in pseudo-labels, pseudo-labels-based methods RegionCLIP and VL-PLM struggle to improve region classification to a higher level. CoOp and CLIP-Adapter suffer from the significant gap between Novel and Base. CoOp learns optimal context tokens for base categories, which are suboptimal for novel categories. Similarly, CLIP-Adapter learns optimal feature transformation for base categories, which are suboptimal for novel categories. Both the context tokens and the linear feature transformation are fixed, facing different objects, resulting in the overfitting of base categories. CORA learns region prompts from base categories and adds the prompts to the visual feature map, which has reduced the performance gap between novel and base categories, but this gap still exists.

To further demonstrate that SIA effectively improves the classification accuracy for regions with extreme aspect ratios, we report the classification performance of regions with different shapes in Figure 4. As shown in the figure, the original CLIP exhibits unsatisfactory classification accuracy for all shapes. CORA improves the overall performance of region classification, but there is still an

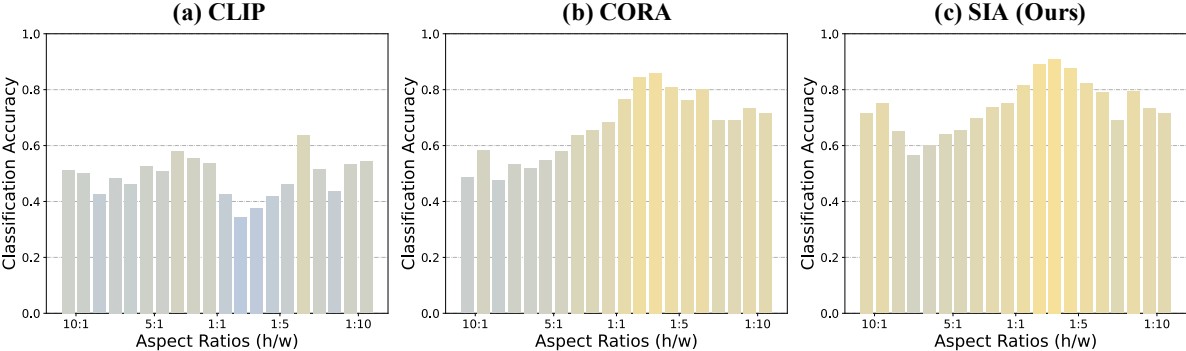

**Figure 4: Classification accuracy for regions with different shapes of CLIP, CORA, and our SIA with RN50 backbone on COCO-OVD validation set with ground-truth bounding boxes.**

**Table 3: Detection performance (i.e., AP50) of each target category on COCO-OVD validation set.**

| Method | Airplane | Bus | Cat | Dog | Cow | Elephant | Umbrella | Tie | Snowboard |
|--------|----------|-----|-----|-----|-----|----------|----------|-----|-----------|
| CLIP | 92.44 | 72.13 | 84.89 | 75.03 | 70.76 | 89.97 | 63.29 | 63.58 | 12.19 |
| CORA | 92.34 | 80.96 | 86.04 | 79.17 | 73.86 | 91.40 | 69.43 | 74.51 | 26.66 |
| SIA | 94.95 | 83.39 | 84.79 | 81.29 | 75.49 | 93.56 | 68.06 | 82.02 | 31.20 |
| Δ | **+2.61** | **+2.43** | **-1.25** | **+2.12** | **+1.63** | **+2.16** | **-1.37** | **+7.51** | **+4.54** |

| Method | Skateboard | Cup | Knife | Cake | Couch | Keyboard | Sink | Sicssors |
|--------|-----------|-----|-------|------|-------|----------|------|----------|
| CLIP | 40.99 | 56.99 | 18.98 | 67.37 | 37.11 | 62.16 | 56.57 | 16.80 |
| CORA | 70.00 | 65.70 | 28.51 | 73.12 | 54.34 | 62.32 | 61.75 | 14.76 |
| SIA | 77.69 | 71.25 | 28.85 | 73.03 | 58.51 | 75.58 | 62.46 | 18.16 |
| Δ | **+7.69** | **+5.55** | **+0.34** | **-0.09** | **+4.17** | **+13.26** | **+0.71** | **+3.40** |

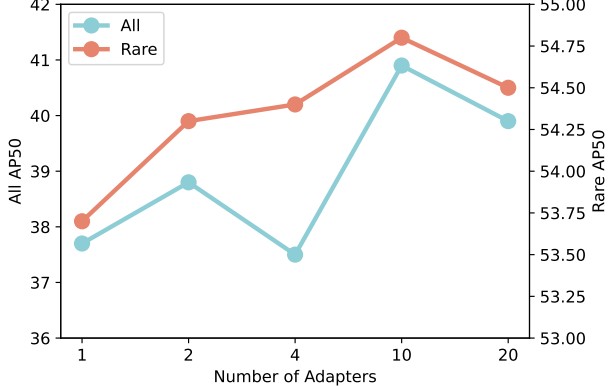

**Figure 5: Effect of the number of adapters on detection performance (AP50) for all and rare categories in LVIS.**

obvious gap between regions with aspect ratios far smaller or larger than 1:1 and regions with aspect ratios close to 1:1. Our proposed approach, SIA, effectively enhances the classification performance of regions with extreme aspect ratios. We also report AP50 scores achieved by CLIP, CORA, and SIA (Ours) across each novel category, as illustrated in Table 3, where $\delta$ denotes the difference between AP50 scores of SIA and CORA. SIA achieves improvements over CORA across most categories, particularly for tie (+7.51%), skateboard (+7.69%), and keyboard (+13.26%). For cat, umbrella, and cake categories, SIA exhibits slight decreases compared to CORA.

**Ablation Study.** We conduct ablation studies of AP50 with ground-truth bounding boxes as input and RN50 as the backbone on the

OV-LVIS benchmark. Figure 5 shows the effect of adapter numbers within SIA architecture. The numbers are selected as 1, 2, 4, 10, and 20. As illustrated in Figure 3, the AP50 initially increases and then decreases as the adapter number increases, reaching its optimal value at 10 adapters. This phenomenon arises because, with fewer adapters, each adapter needs to handle regions with significantly variant shapes, resulting in a relatively severe misalignment between regions and texts. Conversely, with more adapters, each adapter only has a limited number of training samples, especially for regions with extreme aspect ratios.

### 4.4 Visualization.

**Classification Results for Region Proposals.** Figure 6 shows the region classification results in the COCO-OVD validation set. SIA predicts more reasonable classification results than CLIP and CORA for region proposals, especially for regions containing severely deformed objects, e.g., surfboards, ties, and knives. For example, in the first column, SIA successfully identifies the surfboard, while CLIP and CORA misclassify it as an airplane.

**Distribution of Adapted Region Features** Figure 7 shows the region features adapted by SIA for 17 novel categories in COCO-OVD validation set using t-SNE [23]. It is illustrated that SIA achieves a more obvious separation of embeddings belonging to different categories than CLIP and CORA. In (a) CLIP and (b) CORA, feature points of different categories near the center tend to intermingle with each other, making it hard to classify these regions. In (c) SIA, there are relatively more distinct boundaries among feature points of different categories near the center, which verifies that SIA achieves higher classification accuracy for novel categories.

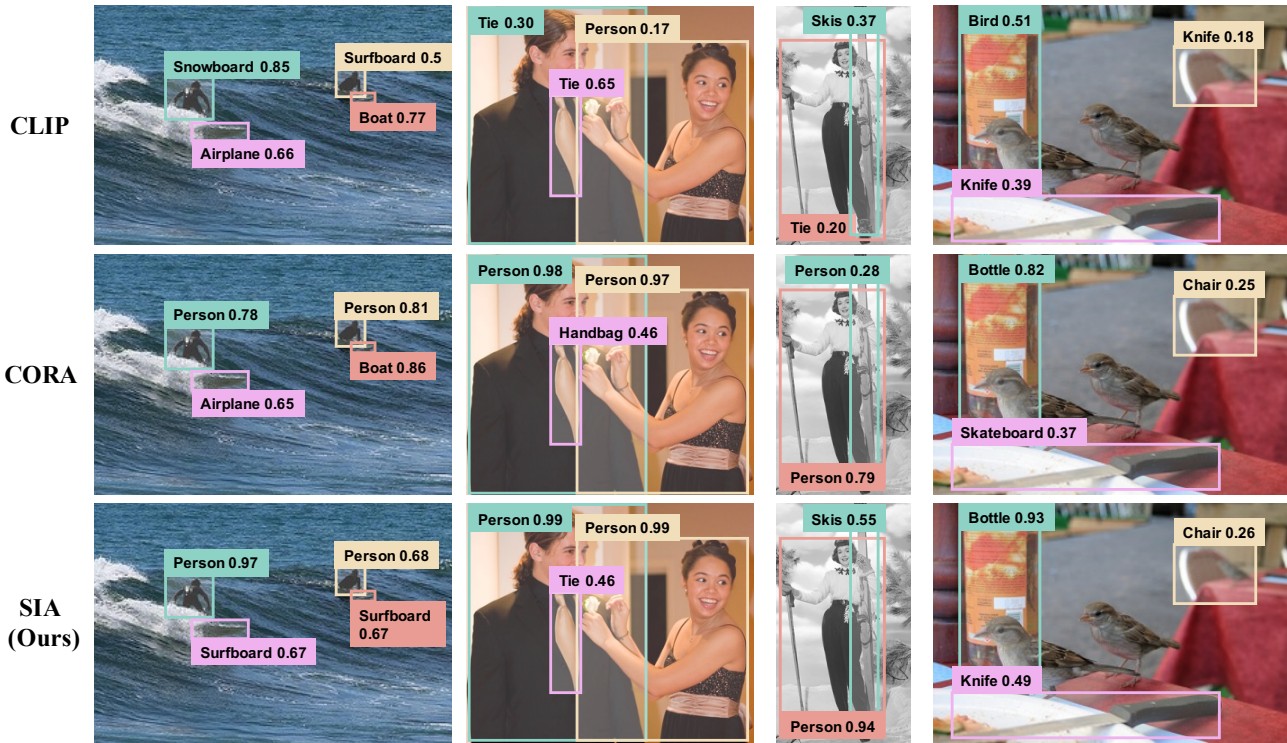

**Figure 6: Visualization of region classification results and confidence scores for ground-truth bounding boxes from COCO-OVD validation set. (Image IDs: 45090, 11051, 52591, 277051)**

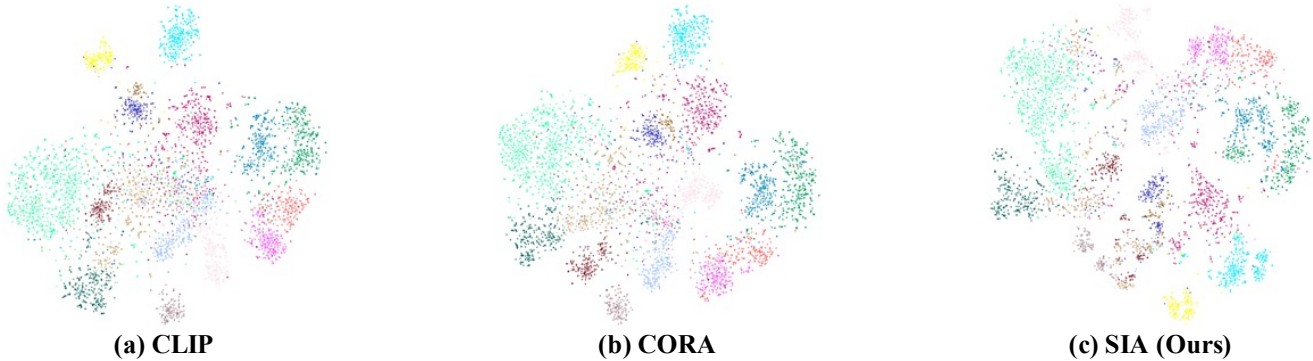

**Figure 7: Visualization of the region features belong to 17 novel categories in the COCO-OVD validation set encoded by CLIP, CORA, and our SIA with RN50 backbone via t-SNE.**

## 5   Conclusion

In this paper, we identify one of the fundamental challenges currently faced by the OVD task: the low region classification accuracy resulting from the gap between the image and the local region, primarily attributed to the shape deformation of region feature maps after RoIAlign cropping operations. To tackle this issue, we propose the Shape-Invariant Adaptor (SIA), which aims to mitigate the misalignment between region features and CLIP's image/text features. SIA maintains a set of feature adapters and assigns different adapters to region proposals with varying shapes through the adapter allocation mechanism. Experimental results demonstrate that SIA effectively enhances the accuracy of region classification,

particularly for regions with deformed object shapes. Our SIA has the potential to inspire and support further research in the OVD field.

**Limitations & Future Work.** SIA focuses on the problem of region classification in the OVD task. However, existing OVD methods still encounter the challenge of localizing novel objects. According to our experimental observations, the RPN network trained on only base categories often fails to accurately and comprehensively generate bounding boxes for objects belonging to novel categories. Therefore, addressing the localization problem of novel categories and exploring one-stage open-vocabulary detectors show significant value for future research.

## Acknowledgments

This work was supported by grants from the National Natural Science Foundation of China (U22B2048, 61925201, 62132001, 62373043).

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
