# OpenReview forum: "SIA-OVD: Shape-Invariant Adapter for Bridging the Image-Region Gap in Open-Vocabulary Detection"
_acmmm.org/ACMMM/2024/Conference — MM2024 Poster_

### Official Review · Reviewer_faU6 · 2024-05-02

**Rating:** 2
**Confidence:** 4

**Summary:**

This paper proposes a new Shape-Invariant Adapter named SIA-OVD to bridge the image-region gap in the OVD task. SIA-OVD learns a set of feature adapters for regions with different shapes and designs a new adapter allocation mechanism to select the optimal adapter for each region. The adapted region representations can align better with text representations learned by CLIP. Extensive experiments demonstrate the effectiveness of SIA-OVD.

**Strengths:**

1.	The paper is with clear structure and easy to understand.

2.	The viewpoint that object shape deformation brought by RoIAlign is the main cause of the image-region gap is meaningful and interesting.

**Limitations:**

1.	What is the difference between RoIAlign for close-set object detection and open-vocabulary object detection? Why do gaps in image regions cause detection obstacles? The introduction did not provide a detailed explanation of Figure 1. Also, the method section is short.

2.	The model is built based on CORA, but the improvement in novel categories in Table 1 is very limited  (RN50 + 0.4AP, RN50 * 4 + 0.2AP). Also, the performance advantage over the main counterpart (CORA) is insignificant.

3.	There seems to be no regularity in the impact of the number of adapters on the performance of base and novel categories. There will be performance fluctuations every time training without changing any structure.

4.	As a plug-and-play module, SIA should be tested on multiple baselines to demonstrate its effectiveness.

5.	Lack of results on the LVIS dataset and comparison with the latest SOTA methods, such as EdaDet [1] and OADP[2].
[1] EdaDet: Open-Vocabulary Object Detection Using Early Dense Alignment. ICCV 2023
[2] Object-Aware Distillation Pyramid for Open-Vocabulary Object Detection. CVPR 2023

6.  This paper focuses on the shape-invariant feature presentation, however, from figure 1 and figure 6, this paper tends to find some tiny object.

**Suitability:**

2

---

### Official Review · Reviewer_cv5z · 2024-05-16

**Rating:** 4
**Confidence:** 4

**Summary:**

This paper is about Open Vocabulary Detection (OVD), which aims to detect novel objects without instance-level annotations, enabling open-world object detection at low cost.

**Strengths:**

（1）This paper has some novelty, summarizes the problem of OAD nowadays, which is the existence of gaps between image regions, and proposes shape invariant adapters to choose different adapters according to the shape of the picture, and the experimental results prove the validity of the proposed module.
（2）The paper is well structured, the figures explain well how this thesis differs from conventional work, and the visualization is adequate and demonstrates well the validity of the proposed methodology of the thesis.

**Limitations:**

（1）In figure2, 'Overview of the OVD framework' or 'Overview of the SIA-OVD framework'?
（2）Insufficient experiments, no ablation experiments of the proposed module are given to carry out the validation of the module's effect.
（3）In the experimental section, comparative results such as the number of parameters and inference speed should be provided。

**Suitability:**

3

---

### Official Review · Reviewer_NDvk · 2024-05-22

**Rating:** 4
**Confidence:** 2

**Summary:**

This paper analyzes and addresses the image-region gap in open-vocabulary detection (OVD) tasks, a fundamental issue caused by object shape deformation induced by the RoIAlign operation. To tackle this problem, the paper introduces a set of learnable shape-invariant adapters (SIAs) designed to map region feature space to a common space. Additionally, an aspect-ratio-related adapter allocation mechanism is developed to handle various shapes effectively. Experimental results on the COCO-OVD benchmark demonstrate the efficiency of the proposed method.

**Strengths:**

This paper analyzes the fundamental image-region gap issue in open-vocabulary detection tasks, underscoring its novelty. To address this issue, the paper introduces shape-invariant adapters and an adapter allocation mechanism, providing an efficient and feasible solution. Moreover, the paper is well-written and easy to follow, clearly elucidating its novelty, methodology, and experiments.

**Limitations:**

Although this paper is well-written, there is still room for improvement.

- Mistake in Eq. (4): $y^{(K)}$
- Unuse notation in line 499: $f_{A}$

Some questions:
- In Sec. 3.2, the paper divides the real number axis of the aspect ratio into $N$ intervals. The reviewer would like to know the details of this division. For instance, is each interval $[s_i, s_{i+1}]$, (i=0,1,...,N-2) of equal length? The reviewer suggests that the division process might be improved by considering the distribution of aspect ratios, which could result in an uneven division.
- It would be better if the authors could make an explanation of the **bolded** and underlined metrics in tables. For example, in Table 1, with the RN50x4 backbone, there is no bolded metric, only an underlined metric.
- The paper conducts experiments only on the COCO-OVD benchmark. The reviewer thinks it would better substantiate the efficacy of the proposed method if the authors could provide additional results on other benchmarks, such as LVIS [1]. Since this paper addresses a fundamental issue in OVD tasks, providing sufficient experimental results would further strengthen the contribution of this work. LVIS benchmark is also considered in recent methods CORA [2] and BARON [3].

[1] Gupta, Agrim, Piotr Dollar, and Ross Girshick. "Lvis: A dataset for large vocabulary instance segmentation." In the IEEE Conference on Computer Vision and Pattern Recognition, pp. 5356-5364. 2019.

[2] Wu, Xiaoshi, Feng Zhu, Rui Zhao, and Hongsheng Li. "Cora: Adapting clip for open-vocabulary detection with region prompting and anchor pre-matching." In the IEEE Conference on Computer Vision and Pattern Recognition, pp. 7031-7040. 2023.

[3] Wu, Size, Wenwei Zhang, Sheng Jin, Wentao Liu, and Chen Change Loy. "Aligning bag of regions for open-vocabulary object detection." In the IEEE Conference on Computer Vision and Pattern Recognition, pp. 15254-15264. 2023.

**Suitability:**

3

---

### Meta-Review · Area_Chair_ihU4 · 2024-07-01

**Recommendation:** Accept (Poster)
**Confidence:** 4

**Metareview:**

This paper analyzes and addresses the image-region gap in open-vocabulary detection (OVD) tasks. There exists an issue caused by object shape deformation induced by the RoIAlign operation. To tackle this problem, the paper introduces a set of learnable shape-invariant adapters (SIAs) designed to map region feature space to a common space. Additionally, an aspect-ratio-related adapter allocation mechanism is developed to handle various shapes effectively. Experimental results on the COCO-OVD benchmark demonstrate the efficiency of the proposed method.

The authors solved most reviewers' concerns in their rebuttal. After the rebuttal, the paper received 2 borderline accept and 1 weak accept.